# Exploration of the Common Gene Characteristics and Molecular Mechanism of Parkinson’s Disease and Crohn’s Disease from Transcriptome Data

**DOI:** 10.3390/brainsci12060774

**Published:** 2022-06-13

**Authors:** Haoran Zheng, Xiaohang Qian, Wotu Tian, Li Cao

**Affiliations:** 1School of Medicine, Anhui University of Science and Technology, Huainan 232001, China; zhenghr@rjlab.cn; 2Department of Neurology Shanghai Jiao Tong University Affiliated Sixth People’s Hospital, Shanghai 200233, China; wotu_tian@163.com; 3Department of Neurology and Institute of Neurology, Rui Jin Hospital, Shanghai Jiao Tong University School of Medicine, Shanghai 200025, China; qianxh@rjlab.cn

**Keywords:** Parkinson’s disease, Crohn’s disease, inflammatory pathway, LYN, RAF1

## Abstract

Parkinson’s disease (PD) is the second most common neurodegenerative disorder, and the mechanism of its occurrence is still not fully elucidated. Accumulating evidence has suggested that the gut acts as a potential origin of PD pathogenesis. Recent studies have identified that inflammatory bowel disease acts as a risk factor for Parkinson’s disease, although the underlying mechanisms remain elusive. The aim of this study was to further explore the molecular mechanism between PD and Crohn’s disease (CD). The gene expression profiles of PD (GSE6613) and CD (GSE119600) were downloaded from the Gene Expression Omnibus (GEO) database and were identified as the common differentially expressed genes (DEGs) between the two diseases. Next, analyses were performed, including functional enrichment analysis, a protein–protein interaction network, core genes identification, and clinical correlation analysis. As a result, 178 common DEGs (113 upregulated genes and 65 downregulated genes) were found between PD and CD. The functional analysis found that they were enriched in regulated exocytosis, immune response, and lipid binding. Twelve essential hub genes including BUB1B, BUB3, DLGAP5, AURKC, CBL, PCNA, RAF1, LYN, RPL39L, MRPL13, RSL24D1, and MRPS11 were identified from the PPI network by using cytoHubba. In addition, inflammatory and metabolic pathways were jointly involved in these two diseases. After verifying expression levels in an independent dataset (GSE99039), a correlation analysis with clinical features showed that LYN and RAF1 genes were associated with the severity of PD. In conclusion, our study revealed the common pathogenesis of PD and CD. These common pathways and hub genes may provide novel insights for mechanism research.

## 1. Introduction

Parkinson’s disease is the second most common neurodegenerative disorder and the fastest-growing neurologic disease globally. It is characterized by the selective degeneration of dopaminergic neurons in the substantia nigra (SN), resulting in dopaminergic depletion in the striatum and the appearance of Lewy bodies harboring α-synuclein aggregates [1,2]. These pathological traits result in cardinal motor symptoms of bradykinesia, rigidity, postural instability, and static tremor, as well as non-motor manifestations, such as sleep disturbances, depression, hyposmia, and intestinal dysfunction [3,4]. Importantly, constipation can occur decades before being diagnosed [5,6]. To date, the mechanisms of PD involve a complex interplay of environmental, age-related, and genetic factors [7]. The gut-brain axis acts as a bridge between the gut and brain, leading to the hypothesis that chronic intestinal inflammation may contribute to PD neurodegeneration and play a role in two diseases [8,9,10]. In addition, the gut-origin hypothesis implicates the gut as a potential origin of PD pathogenesis, providing fresh insights into the mechanisms underlying PD and IBD [11]. Intestinal dysbiosis and Lewy body formation have been established in PD patients in an increasing number of investigations. In recent years, various digestive illnesses, particularly IBD, have been identified as major risk factors for PD [12]. IBD, which includes Crohn’s disease (CD) and ulcerative colitis (UC), is a chronic proinflammatory immunological disorder that develops in young adulthood [13]. However, the potential relationship between these two chronic progressive diseases remains unclarified.

It sparked a heated debate concerning the association between PD and IBD. However, data from studies showed no conclusive evidence supporting CD as a risk factor for PD [14]. Cohort studies in Danish and Taiwanese demonstrated that CD patients had a relatively higher risk of PD [15,16]. In addition, CD was also linked to an increased risk of PD in two other studies [2,17]. Furthermore, past research revealed that chronic systemic inflammation increases the risk of PD. Some peripheral inflammatory cytokines such as tumor necrosis factor (TNF), IL-2 IL-1β, and IL-10 are elevated in PD patients [18,19,20]. CD has also been proven in an increasing number of clinical investigations to raise the likelihood of developing PD and to encourage the development of PD [2,13,16,20]. Meanwhile, new research revealed that CD patients on immunosuppressive therapy can lessen the risk of IBD-related PD [12,14,21].

Over 90 risk variants have been identified in genome-wide association studies (GWAS), and common genetic variants such as Nucleotide-bling oligomerization domain2 (NOD2), leucine-rich repeat kinase 2 (LRRK2), and microtubule-associated protein tau (MAPT) genes have been found between PD and CD [7,20]. These mutations are involved in regulating autoimmune and inflammatory diseases, which have been identified as significant risk factors for PD [2,7]. Despite the fact that CD is thought to be a risk factor for PD, the exact processes that explain the coexistence of these two diseases are unknown.

In this study, we obtained the peripheral blood transcriptome expression data (GSE6613 and GSE99039) of CD and PD patients from the GEO database to explore the common molecular mechanism between the two diseases. In addition, protein–protein interaction (PPI) nodes were constructed to analyze gene modules and identify hub genes by using the search tool for the retrieval of the interacting genes/proteins (STRING) database and Cytoscape software, respectively. To improve the reliability of the results, we verified the expression level of hub genes and analyzed the correlation of PD clinical characteristics in independent datasets (GSE99039). This study will provide new insights into the potential pathogenesis mechanism of PD and CD. 

## 2. Materials and Methods

### 2.1. Data Collection and Processing

The peripheral blood RNA transcriptome profile of patients with PD and CD was downloaded from the NCBI Gene Expression Omnibus public database (GEO, https://www.ncbi.nlm.nih.gov/geo/ (accessed on 28 March 2022)). Two datasets (GSE6613 and GSE99039) of Parkinson’s disease were included in this study. Among them, 22 healthy controls and 55 Parkinson’s disease patients in GSE6613 were used as the test data set. The GSE99039, including 233 healthy controls and 205 Parkinson’s disease patients, was used as the verification dataset. The 47 adult healthy controls and 48 adult Parkinson’s disease patients in GSE119600 were used to screen for different genes associated with Crohn’s disease.

### 2.2. Identification of Differentially Expressed Genes

The gene expression profiles of differentially expressed genes (DEGs) between the PD patients or CD patients and HC was identified through the Limma package in R3.4.1 [22]. Genes with |log2FC (fold change)| > 0, and *p* value  <  0.05, were considered to be DEGs. DEGs were shown in the volcano map. The 30 genes with the most significant differences in up-regulation and down-regulation were shown in heat maps. 

### 2.3. Functional Enrichment Analysis of Significant DEGs

To further clarify the potential functional annotation and pathway enrichment associated with the DEGs and hub gene, Gene Ontology (GO) analyses, including cellular component (CC), biological process (BP), molecular function (MF), and Kyoto Encyclopedia of Genes and Genomes (KEGG) pathways were performed to figure out the functional roles of Robust DEGs by the clusterProfiler package [23]. Statistical significance was defined as a *p* value < 0.05.

### 2.4. Protein–Protein Interaction (PPI) Network and Module Analysis

The STRING online tool (https://cn.string-db.org/) (accessed on 28 March 2022) was used to construct the PPI network with the threshold of the combined score > 0.4 [24]. The PPI network was visualized using the Cytoscape application [25]. The Molecular complex detection (MCODE) and CytoHubba plugin of Cytoscape were applied to screen out the significant modules and core genes, respectively. Three different algorithms (Maximum Neighborhood Component (MNC), Maximal Clique Centrality (MCC), and Edge Percolated Component (EPC)) were used for hub genes screening. Finally, the common genes obtained by the three algorithms were identified as reliable hub genes and demonstrated using a Venn diagram. 

### 2.5. Statistical Analysis

Statistical analysis and graphs were performed using R software. *p* value < 0.05 was considered statistically significant.

## 3. Results

### 3.1. Identification of Common DEGs between PD and CD

The research flowchart of this study was shown in Figure 1. Samples from two datasets, GSE6613 and GSE99039, were downloaded from the GEO database according to the previous methodologies and criteria. A differential gene analysis was carried out. Compared to the controls, we found 841 genes (455 upregulated and 386 downregulated genes) as significant DEGs in patients with PD, and 10,364 genes (5500 upregulated and 4864 downregulated genes) (Figure 2A,C) were identified as DEGs in CD. Meanwhile, heat maps depicted the top 20 DEGs in both diseases as a result of the cluster analysis (Figure 2B,D). The intersection of the Venn diagram yielded 178 communal DEGs (Figure 2E,F), including 113 co-upregulated genes and 65 co-downregulated genes.

### 3.2. Analysis of the Functional Characteristics of Common DEGs

To figure out the function and pathways of 178 communal DEGs between PD and CD, “clusterprofiler” was used for GO and KEGG pathway enrichment. In terms of the biological process, cellular component, and molecular function results are presented in Figure 3. These genes were found to be concentrated in the cytosol, endomembrane system, and vesicle in terms of cellular components (Figure 3A). A molecular function analysis demonstrated that these genes enriched in lipid binding, enzyme activator activity, and GTPase regulator activity (Figure 3B). Biological process outcomes revealed that these genes enriched in the regulated exocytosis, leukocyte activation involved in immune response, and cell activation involved in immune response (Figure 3C). Furthermore, a KEGG pathway enrichment analysis revealed that Pathogenic Escherichia coli infection, Cholesterol metabolism, Apoptosis, MAPK signaling pathway, NF-kappa B (NF-kb) signaling pathway, and PI3K-Akt signaling pathway were heavily enriched. (Figure 3D). These findings indicated that inflammatory pathways and metabolism pathways are implicated in the progression of these two chronic progressive diseases. 

### 3.3. PPI Network Analysis and Identification of Hub Gene

The STRING database was used to perform a PPI network analysis of the communal DEGs to clarify the interactions between DEGs (Figure 4A). Here, we further explored potential small modules through the MCODE plug-in of Cytoscape, of which there are four modules, including 15 common DEGs (Figure 4B–E). We further identified the top 45 hub genes in the public DEGs (Figure 5A–C) by using the three algorithms MCC, EPC, and MNC, respectively, in the cytoHubba plugin. Finally, the three algorithms yielded a total of 12 common hub genes, including BUB1B, BUB3, DLGAP5, AURKC, CBL, PCNA, RAF1, LYN, RPL39L, MRPL13, RSL24D1, and MRPS11 in the Venn diagrams (Figure 5D). In addition, we performed a GO and KEGG enrichment analysis on the 12 hub genes. The GO analysis showed that these genes are primarily involved in the peptide metabolic process, protein phosphorylation, microtubule-binding, structural molecule activity, plasma membrane region, and mitochondrion (Figure 6A–C). The importance of the metabolic process in these two diseases was underscored by these findings. Meanwhile, the KEGG pathway analysis revealed that these genes were primarily involved in the cell cycle, ribosome, chemokine signaling pathway, NF-kappa B (NF-kb) signaling pathway, and Fc epsilon RI signaling pathway (Figure 6D). The functional annotation of 12 hub genes matches the results of 178 common DEGs, lending credence to the idea of metabolic processes and inflammatory pathways acting as the common pathological mechanism of PD and CD.

### 3.4. Validation of Hub Genes and Correlation Analysis with Clinical Features

The expression level of hub genes was verified in 233 healthy controls and 205 PD patients in order to improve the dependability of hub genes (Figure 7A). Consistent with the findings of this study, LYN and RAF1 were significantly overexpressed in the PD group. The remaining ten hub genes showed no significant differences. The Unified Parkinson’s Disease Rating Scale (UPDRS) (Figure 7B–I) of PD patients provides critical indicators to evaluate the clinical severity of PD patients [26,27]. The LYN expression levels were found to be positively correlated with UPDRS I (r = 0.166, *p* = 0.003); UPDRS II (r = 0.236, *p* < 0.01); UPDRSIII (r = 0.203, *p* < 0.01); UPDRS IV (r = 0.269, *p* < 0.01). The RAF1 expression levels were found to be positively correlated with UPDRS I (r = 0.075, *p* = 0.18); UPDRS II (r = 0.118, *p* = 0.036); UPDRSIII (r = 0.145, *p* = 0.014); UPDRS IV (r = 0.109, *p* = 0.050), implying that LYN and RAF1 are directly linked to the severity of PD.

## 4. Discussion

Parkinson’s disease is a chronic age-related neurodegenerative disease. The main symptoms of PD are severe motor disturbances, including tremors, postural imbalance, slow movement, and rigidity, and non-motor symptoms such as depression, hyposmia, and constipation. A growing body of epidemiological evidence suggested that IBD patients have a significantly increased risk of PD. According to a meta-analysis, IBD conferred a 28–30% increased risk of PD [14,28]. However, the current understanding of the molecular mechanisms of how IBD increases the risk of developing PD is still very limited. Based on the minimal research available, Leucine-rich-repeat kinase 2 (LRRK2) is thought to be a key genetic link between PD and IBD. LRRK2, with kinase and GTPase activity, is abundant in neurons, glial cells, and peripheral immune cells [29,30,31]. Numerous studies have demonstrated that LRRK2 is involved in protein synthesis, immune response regulation, inflammation, and other cellular functions [32]. Increased LRRK2 activity may enhance the sensitivity of gut inflammation and generate systemic inflammation, both of which can lead to the development of PD [12]. A study reported that CD patients with LRRK2 (p.N2081D and p.G2019S) are more likely to develop PD [7,32,33]. In this study, we identified 178 common DEGs involved in the chemokine signaling pathway, MAPK signaling pathway, NF-kappa B (NF-kb) signaling pathway, and Fc epsilon RI signaling pathway from whole peripheral blood between PD and CD. We further identified 12 core genes from the above common DEGs, including BUB1B, BUB3, DLGAP5, AURKC, CBL, PCNA, RAF1, LYN, RPL39L, MRPL13, RSL24D1, and MRPS11. These genes were significantly enriched in the inflammatory and metabolic pathways including cell cycle, apoptosis, and peptide metabolic process, according to the GO and KEGG pathway, and enrichment analysis. Hereby, we hypothesize that the inflammatory and metabolism pathways play an important role as public pathways in PD and CD. Inflammatory and immunological regulations including chemokines and cytokines, such as TNF-, IL-17, IL-6, and IL-2, are mutually implicated in the development of these two inflammatory disorders [7]. α-synuclein is a key participant in PD-associated inflammation, inducing particular T-cell activity and stimulating microglial activation in the central nervous system (CNS), which is engaged in the pathogenic processes of PD disease [34]. Meanwhile, these immune mediators have been linked to the Nuclear Factor Kappa B (NF-kB) signaling pathway, which is involved in lipid metabolism and regulation of immune response. According to KEGG analysis, MAPK, NF-kB, and PI3K-Akt signaling pathways are widely used and involved in various physiological regulation processes, including lipid metabolism, and the regulation of immune response. The PI3K-Akt signaling pathways in CD may control the intestinal immune-inflammatory response [35]. In PD, Lipopolysaccharide (LPS) can regulate cytokine release and modulate the immunological response by activating the Toll-like receptor (TLR)-linked signaling pathway and elevating the phosphorylation levels of MAPK and NF-kB pathways [36,37]. In addition, we further verified the expression level of hub genes and found that two genes, including LYN and RAF1, were highly expressed in PD. After that, we validated the expression levels of 12 hub genes using the independent dataset GSE99039 and found that LYN and RAF1 remained high and significantly elevated in the PD group, and their expression levels correlated with clinical scale scores in PD. This suggests that the above two genes play an important role in the procession of PD.

LYN (Lck/yes novel tyrosine kinase) is a member of the Src family with four distinct domains, including SH1, SH2, SH3, and SH4. Lyn is a vital signal intermediary that modulates different processes such as apoptosis, immune response, and metabolism. PD inhibits Lyn kinase activity and downregulates downstream signaling pathways, including MAPK, PI3K/AKT, and NF-kB, which could suppress inflammatory responses [38]. The activity status of LYN can exert effects on immune cells and regulate the immune response, which is associated with PD. A previous study showed that LYN could sense changes in H_2_O_2_ concentration in microglia and phosphorylate them, resulting in α-syn–mediated microglial migration in PD [39]. LYN is also engaged in α-synuclein-induced microglial migration via cytoskeleton remodeling, indicating that LYN regulation might play a significant role in PD [40]. However, more research into the long-term effects of LYN on preclinical PD models is required. In summary, immune cells probably modulate the LYN in different signal pathways, which may mediate the crosstalk of PD. 

RAF1 belongs to the Raf family of serine/threonine protein kinases, consists of A-Raf, B-Raf, and Raf-1 (C-Raf) and has three conserved regions, and all of these proteins play a vital role in the mitogen-activated protein kinases (MAPK) pathway [41,42]. In addition, intestinal fibrosis is the main pathological process in Crohn’s disease. The present study demonstrated that Moxibustion can downregulate the phosphorylation of the Ras, Raf-1, MEK-1, and ERK-1/2 proteins and the expression of the corresponding mRNAs in the colon tissue in CD by regulating the ERK signaling pathway [43]. Some studies demonstrated that the Raf kinase inhibitor protein (RKIP) is involved in the CDK5/RKIP/ERK pathway in PD pathogenesis and provides a potential therapeutic target in PD [44].

Although previous studies explored the hub genes mainly enriched in immune and metabolic processes and that may be involved in MAPK, NF-kB, and PI3K-Akt signaling pathways, the underlying molecular mechanisms have not yet been fully elucidated. In future studies, we will further validate the role of the above core genes and signaling pathways in PD and CD at the cellular and animal levels.

## 5. Conclusions

In summary, abnormal inflammatory and metabolic signaling pathways are common pathogenic pathways between PD and CD. LYN and RAF1 were identified as novel common hub genes of PD and CD, which are involved in disease development by regulating immune pathways.

## Figures and Tables

**Figure 1 brainsci-12-00774-f001:**
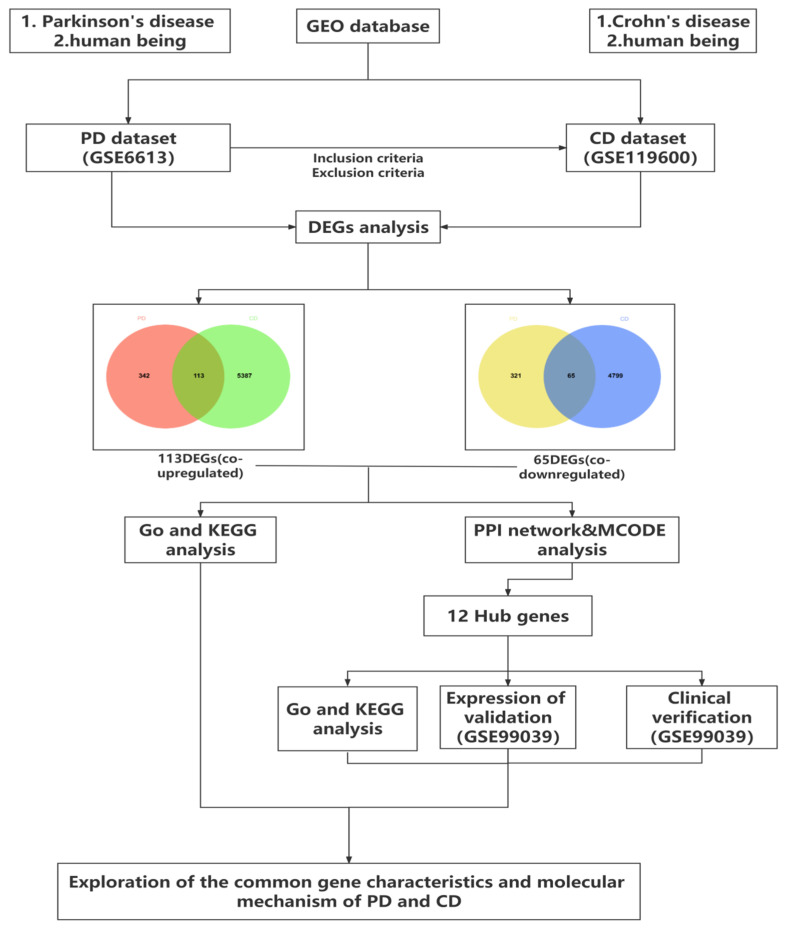
Research design flow chart.

**Figure 2 brainsci-12-00774-f002:**
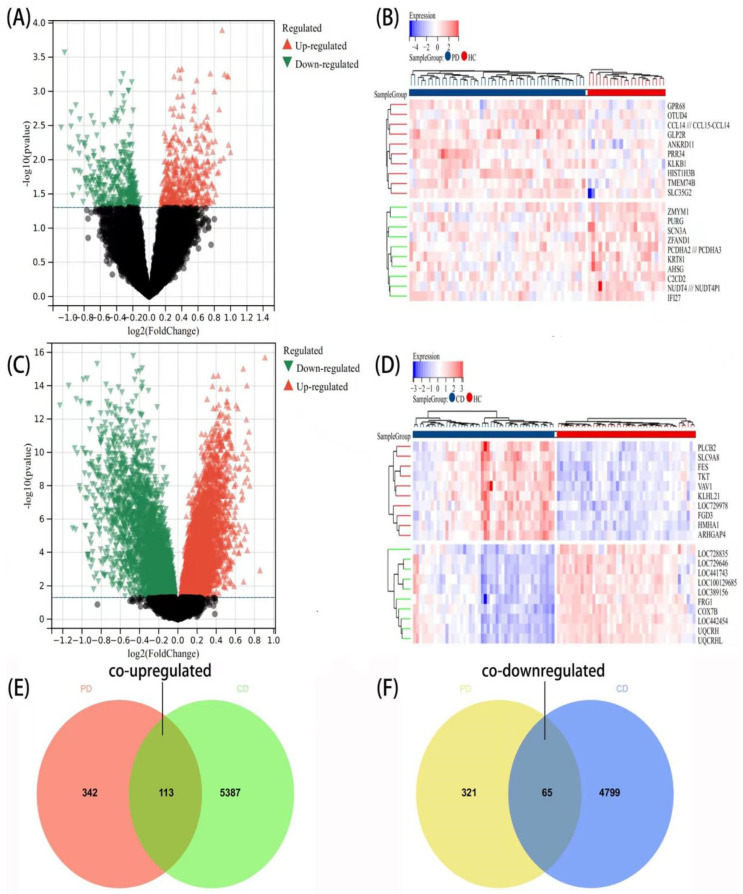
Identification of gene expression profiles in the two datasets. (**A**) The volcano map showed the DEGs between PD and HC group. Upregulated genes are marked in light red; Downregulated genes are marked in light green. The threshold was set to |log2FC (fold change)| > 0, and *p* value  <  0.05. (**B**) The cluster heat map of top 20 DEGs between PD and HC group. (**C**) The volcano map showed the DEGs between CD and HC group. (**D**) The cluster heat map of top 20 DEGs between CD and HC group. (**E**) The Venn diagram showed an overlap of 113 co-upregulated DEGs. (**F**) The Venn diagram indicates an overlap of 65 co-downregulated DEGs.

**Figure 3 brainsci-12-00774-f003:**
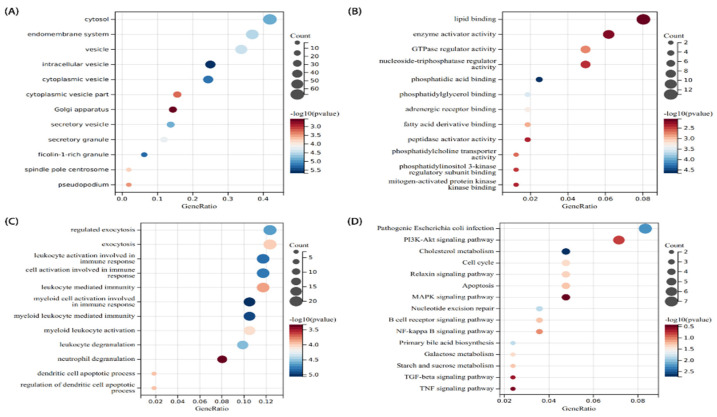
Functional annotation of communal DEGs. (**A**) The cellular component analysis results. (**B**) The molecular function analysis. (**C**) The biological process analysis results. (**D**) The KEGG analysis results. *p*-value  <  0.05 was considered significant.

**Figure 4 brainsci-12-00774-f004:**
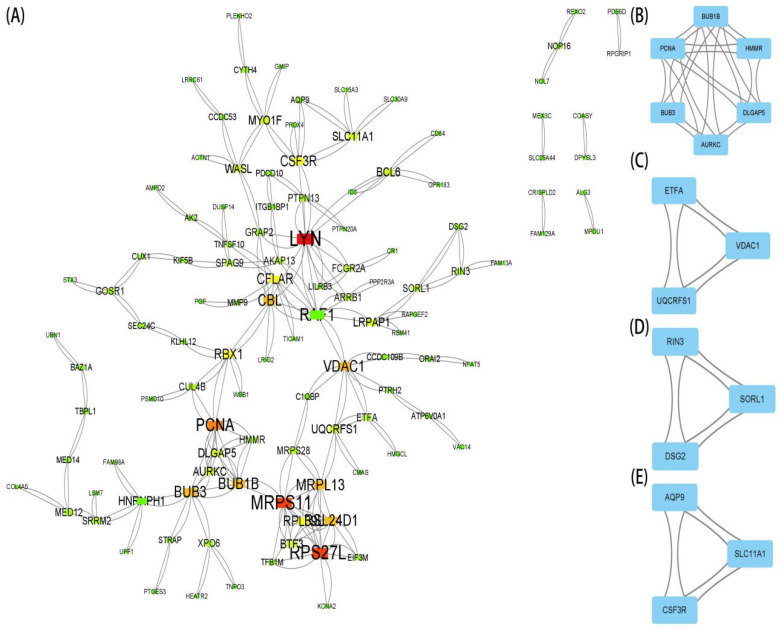
PPI network and significant gene module. (**A**) PPI network diagram. (**B**–**E**) The significant gene clustering modules.

**Figure 5 brainsci-12-00774-f005:**
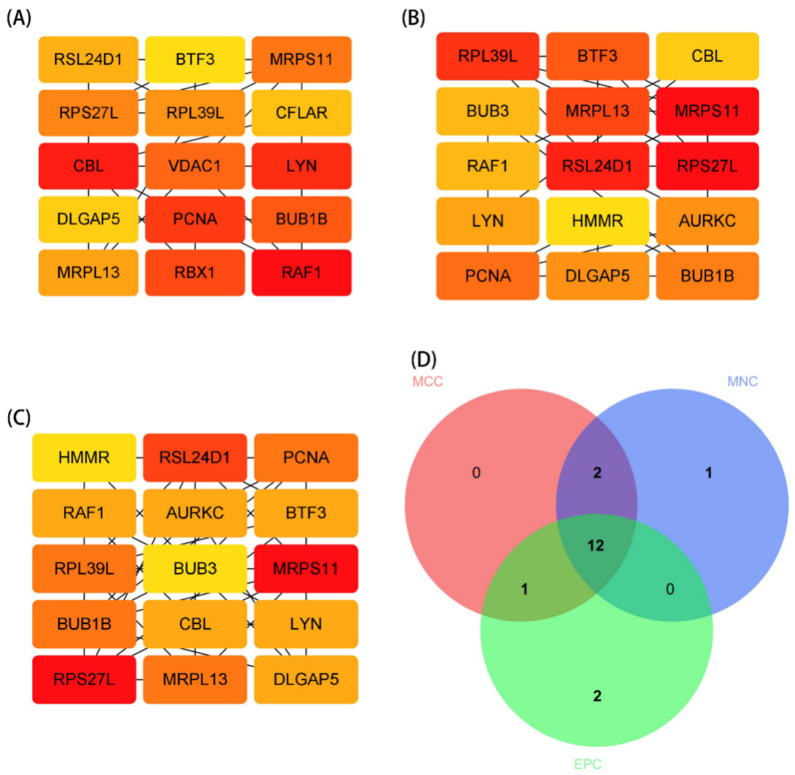
Screening of hub differentially expressed genes (DEGs). (**A**–**C**) Hub DEGs screened by the MCC, EPC, and MNC algorithms. (**D**) The Venn diagram showed the common hub DEGs between MCC, EPC, and MNC algorithms.

**Figure 6 brainsci-12-00774-f006:**
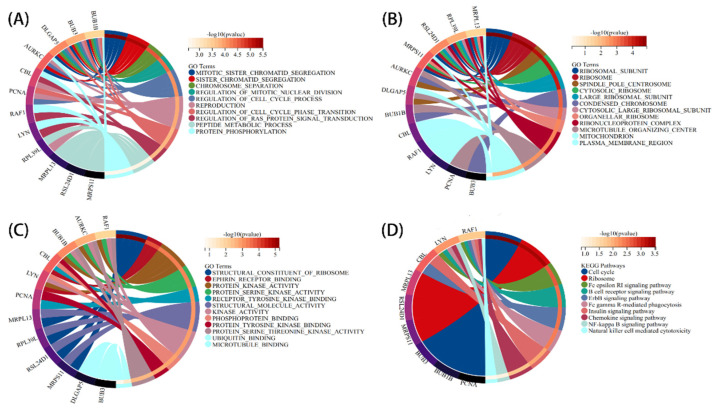
Functional annotation of the hub genes. (**A**–**C**) The GO enrichment analysis of the hub genes. (**D**) The KEGG pathway enrichment analysis of the hub genes.

**Figure 7 brainsci-12-00774-f007:**
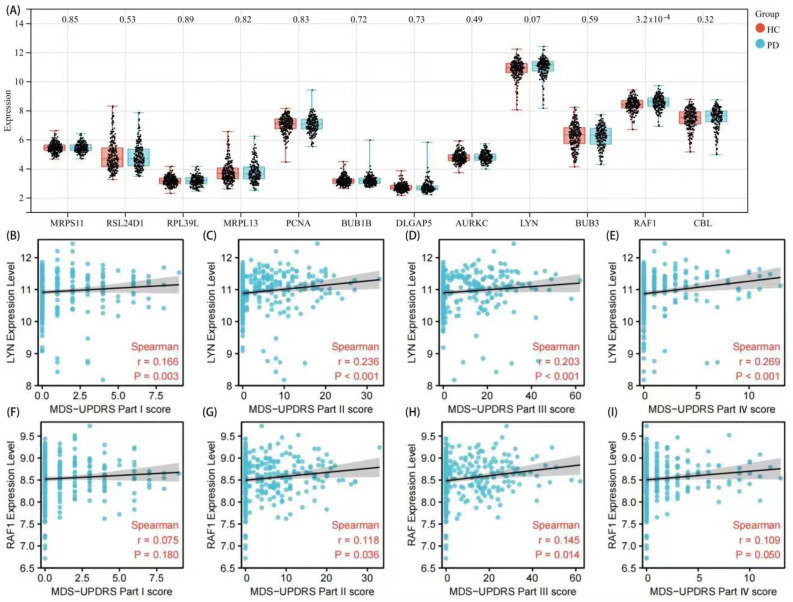
The expression level of hub gene in GSE99039 and clinical verification. (**A**) The expression level of hub genes in GSE99039. (**B**–**E**) The LYN expression level (UPDRS I-IV). (**F**–**I**) The RAF1 expression level (UPDRS I-IV).

## Data Availability

All data related to the research are presented in the article.

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
