# Peer review of "Exploration of the Common Gene Characteristics and Molecular Mechanism of Parkinson’s Disease and Crohn’s Disease from Transcriptome Data"

_brainsci, 2022, doi:10.3390/brainsci12060774_

Round 1

Reviewer 1 Report

The submitted study aimed to evaluate genes from patients with Crohn's disease and Parkinson's disease, as a means to identify shared genes and possible shared pathological pathways. The authors use available datasets and were able to identify multiple shared genes that were upregulated and downregulated in PD and CD. Of these LYN and Raf1 were singled out as of particular interest.

Although the methodological approach that was taken by the authors is solid, the findings do not add a significant amount of new information to the field. The authors have identified the role of inflammation between PD and CD, which has already been shown in numerous other studies. Further, the validation of select genes (Figure 7A) between health control and PD patients does not demonstrate a dramatic change in any of the genes between HC and PD. Further, there does not appear to be a significant associated between LYN and Raf1 and the H&Y or UPDRS. Without these findings, there is not a lot of novel information presented in this submission.

Author Response

Response: We appreciate the suggestion of the reviewer. The inflammatory pathway is a common pathway for both CD and PD as previously reported. In our study, we further explored the molecular mechanism and various inflammatory pathways. We also identified novel genes, such as LYN and RAF1 of PD and CD that were not previously reported. The transcriptional profile of peripheral blood differs from that of the central nervous system. Considering the reviewer’s suggestion, we will continue to explore the potential mechanisms between PD and CD in the future.

Reviewer 2 Report

Review of a manuscript “Exploration of the common gene characteristics and molecular mechanism of Parkinson's disease and Crohn's disease from transcriptome data” by Zheng et al., submitted to “Brain Science”

Accumulating results point to the association between Parkinson’s disease and inflammatory bowel disease, however molecular mechanism linking these two disorders are not completely understood. Inflammatory bowel disease is a chronic proinflammatory immune condition which includes Crohn’s disease and ulcerative colitis.  The authors try to identify genes implicated in the pathogenesis of Parkinson’s disease and Crohn's disease. This is important field of biomedical research, and he results presented in the manuscript will be interesting for the readers of the journal.

The following corrections and additions should be made.

Abstract

“Accumulating evidence has suggested that the gut as a potential origin of PD pathogenesis originates from the gut-origin hypothesis”. This is an awkward sentence which should be truncated as follows: “Accumulating evidence has suggested that the gut as a potential origin of PD pathogenesis.”

Introduction

After the sentence “Parkinson’s disease is the second most common neurodegenerative disorder and the fastest-growing neurologic disease globally” the authors should add a recent review:”Surguchov A. Biomarkers in Parkinson’s Disease”. Chapter in a book Peplow P.V., Martinez B., Gennarelli T.A. (eds) Neurodegenerative Diseases Biomarkers. 2022. Neuromethods, vol 173. pp 155-180. Humana, New York, NY. https://link.springer.com/protocol/10.1007/978-1-0716-1712-0_7

Figure 1. Rearch design flow chart. Corrections required:

1 Rearch should be corrected as Research

2 The fonts are too small, should be increased for easier reading

Figure 2. The text for Figure 2B and 2D should be increased in size for easier reading

Discussion

 “Numerous functional studies for LRRK2 have been reported, including protein synthesis, immune response regulation, inflammation, and among others[32].”Unclear sentence. What the authors want to say? Probably: “Numerous studies have demonstrated that LRRK2 is involved in protein synthesis, immune response regulation, inflammation, and others cellular functions.”

“In addition, the present study showed that the expression of the Ras, Raf-1, MEK-1 and ERK-1/2 proteins were increased in CD intestinal fibrosis by regulating the ERK signaling pathway[41].” The sense of this sentence is unclear. What the authors want to say:  The expression of genes is increased by changing ERK signaling pathway? How expression changes are associated with signaling pathway? Should be clarified.

Author Response

We would like to express our sincere appreciation to the reviewers for the constructive and positive comments.

Replies to Reviewer 2:

Comment 1: Abstract: “Accumulating evidence has suggested that the gut as a potential origin of PD pathogenesis originates from the gut-origin hypothesis”. This is an awkward sentence which should be truncated as follows: “Accumulating evidence has suggested that the gut as a potential origin of PD pathogenesis.”

Response: We appreciate the concern of the reviewer. We rephrased the related sentences with distinctive colors in the abstract, which was also listed below.

Abstract (Line16-17)

Accumulating evidence has suggested that the gut acts as a potential origin of PD pathogenesis.

Comment 2: Introduction: After the sentence “Parkinson’s disease is the second most common neurodegenerative disorder and the fastest-growing neurologic disease globally” the authors should add a recent review: Surguchov A. “Biomarkers in Parkinson’s Disease”. Chapter in a book Peplow P.V., Martinez B., Gennarelli T.A. (eds) Neurodegenerative Diseases Biomarkers. 2022. Neuromethods, vol 173. pp 155-180. Humana, New York, NY. https://link.springer.com/protocol/10.1007/978-1-0716-1712-0_7

Response: We appreciate the suggestion from the reviewer. We added this recent review in the introduction.

Introduction (Line38-41)

It is characterized by the selective degeneration of dopaminergic neurons in the substantia nigra (SN), resulting in dopaminergic depletion in the striatum and the appearance of Lewy bodies harboring α-synuclein aggregates [1,2].

Comment 3: Figure 1: Rearch design flow chart. Corrections required:1 Rearch should be corrected as Research. 2 The fonts are too small, should be increased for easier reading

Response: We appreciate the concern of the reviewer. We revised the typo in Figure 1 and we also adjusted the fonts of Figure1.

Comment 4: Figure 2: The text for Figure 2B and 2D should be increased in size for easier reading

Response: We appreciate the suggestion of the reviewer. We adjusted the size of Figure2B and 2D accordingly.

Comment 5: Discussion: “Numerous functional studies for LRRK2 have been reported, including protein synthesis, immune response regulation, inflammation, and among others [32].” Unclear sentence. What the authors want to say? Probably: “Numerous studies have demonstrated that LRRK2 is involved in protein synthesis, immune response regulation, inflammation, and others cellular functions.”

Response: We appreciate the suggestion of the reviewer. We rephrased this sentence with distinctive colors in the discussion and also listed it below.

Discussion (Line214-216):

Numerous studies have demonstrated that LRRK2 is involved in protein synthesis, immune response regulation, inflammation, and other cellular functions

Comment 6: Discussion: “In addition, the present study showed that the expression of the Ras, Raf-1, MEK-1 and ERK-1/2 proteins were increased in CD intestinal fibrosis by regulating the ERK signaling pathway [41].” The sense of this sentence is unclear. What the authors want to say:  The expression of genes is increased by changing ERK signaling pathway? How expression changes are associated with signaling pathway? Should be clarified.

Response: We appreciate the suggestion of the reviewer. We revised this sentence with distinctive colors in the discussion.

Discussion (Line 263-267)

In addition, intestinal fibrosis is the main pathological process in Crohn's disease. The present study showed that Moxibustion can downregulate the phosphorylation of the Ras, Raf-1, MEK-1, and ERK-1/2 proteins and the expression of the corresponding mRNAs in the colon tissue in CD by regulating the ERK signaling pathway [43].

Reviewer 3 Report

The manuscript “Exploration of the common gene characteristics and molecular mechanism of Parkinson's disease and Crohn's disease from transcriptome data” by Zheng et al explores the common genes between Parkinson's disease and Crohn's disease. The study is an interesting read, however, a few improvements must be included to improve the overall quality of the manuscript

Please include a clear image of the volcano and a heat map – Figure-2

Similarly for figures 4, and 7, please provide a clear and readable image

Discussion - The authors have mentioned some of the common genes were either upregulated or downregulated in their respective disease domains (i.e PD and CD). Please elaborate on how these differentially expressed genes contribute from one disease to another disease pathology for better understanding.

Author Response

We would like to express our sincere appreciation to the reviewers for the constructive and positive comments.

Replies to Reviewer 3:

Comment 1: Please include a clear image of the volcano and a heat map – Figure-2, Similarly for figures 4, and 7, please provide a clear and readable image.

Response: We appreciate the suggestion of the reviewer. We revised the resolution of Figure 2 (the volcano and a heat map) and Figure4, and rephrased the related figure legends in Figure 2 and 7.

Figure2. (A) The volcano map showed the DEGs between PD and HC group. Upregulated genes are marked in light red; Downregulated genes are marked in light green. The threshold was set to |log2FC (fold change) | > 0, and P value < 0.05. (B) The cluster heat map of top 20 DEGs between PD and HC group. (C) The volcano map showed the DEGs between CD and HC group. (D) The cluster heat map of top 20 DEGs between CD and HC group. (E) The venn diagram showed an overlap of 113 co-upregulated DEGs. (F) The venn diagram showed an overlap of 65 co-downregulated DEGs.

Figure7. (B-E) The LYN expression level (UPDRSI-IV). (F-I) The RAF1 expression level (UPDRSI-IV).

Comment 2: Discussion - The authors have mentioned some of the common genes were either upregulated or downregulated in their respective disease domains (i.e PD and CD). Please elaborate on how these differentially expressed genes contribute from one disease to another disease pathology for better understanding.

Response: We appreciate the suggestion of the reviewer. We add related sentences in the discussion.

Discussion (Line230-233)

α-synuclein is a key participant in PD-associated inflammation, inducing particular T-cell activity and stimulating microglial activation in the central nervous system (CNS), which is engaged in the pathogenic processes of PD disease [35].

Discussion (Line256-258)

LYN is also engaged in α-synuclein-induced microglial migration via cytoskeleton remodeling, indicating that LYN regulation might play a significant role in PD [41].

Discussion (Line267-269)

Some studies have shown that Raf kinase inhibitor protein (RKIP) is involved in the CDK5/RKIP/ERK pathway in PD pathogenesis and provides a potential therapeutic target in PD [45].

Round 2

Reviewer 1 Report

Unfortunately, the authors have made no attempt to address the concerns that were orginally expressed.